# Phenotyping the Chemical Communications of the Intestinal Microbiota and the Host: Secondary Bile Acids as Postbiotics

**DOI:** 10.3390/cells14080595

**Published:** 2025-04-15

**Authors:** Ginevra Urbani, Elena Rondini, Eleonora Distrutti, Silvia Marchianò, Michele Biagioli, Stefano Fiorucci

**Affiliations:** 1Dipartimento di Medicina e Chirurgia, Università degli Studi di Perugia, 06123 Perugia, Italy; ginevra.urbani@dottotandi.unipg.it (G.U.); silvia4as@hotmail.it (S.M.); michele.biagioli@unipg.it (M.B.); 2SC di Gastroenterologia ed Epatologia, Azienda Ospedaliera di Perugia, 06123 Perugia, Italy; elena.rondini@specializzandi.unipg.it (E.R.); eleonora.distrutti@ospedale.perugia.it (E.D.)

**Keywords:** postbiotics, secondary bile acids, immune system, human health

## Abstract

The current definition of a postbiotic is a “preparation of inanimate microorganisms and/or their components that confers a health benefit on the host”. Postbiotics can be mainly classified as metabolites, derived from intestinal bacterial fermentation, or structural components, as intrinsic constituents of the microbial cell. Secondary bile acids deoxycholic acid (DCA) and lithocholic acid (LCA) are bacterial metabolites generated by the enzymatic modifications of primary bile acids by microbial enzymes. Secondary bile acids function as receptor ligands modulating the activity of a family of bile-acid-regulated receptors (BARRs), including GPBAR1, Vitamin D (VDR) receptor and RORγT expressed by various cell types within the entire human body. Secondary bile acids integrate the definition of postbiotics, exerting potential beneficial effects on human health given their ability to regulate multiple biological processes such as glucose metabolism, energy expenditure and inflammation/immunity. Although there is evidence that bile acids might be harmful to the intestine, most of this evidence does not account for intestinal dysbiosis. This review examines this novel conceptual framework of secondary bile acids as postbiotics and how these mediators participate in maintaining host health.

## 1. Definition of Postbiotics: Differences Between Prebiotics and Probiotics

According to the International Scientific Association for Probiotics and Prebiotics (ISAPP), a postbiotic is a “preparation of inanimate microorganisms and/or their components that confers a health benefit on the host” [1]. Despite the fact that multiple definitions of “postbiotic” have been previously proposed (Table 1), taking or not in consideration the inclusion of microbial cells in the preparation, none of them resulted in perfectly fitting with the intrinsic meaning of this concept.

Within the definition of a postbiotic, intact microorganisms are not required for health effects: what is needed, as a part of the manufacturing process of a postbiotic, is a deliberate chemical or physical process of viability termination (via heat, radiation, high pressure or lysis) that may maintain (or not) cell integrity [8]. To qualify as a postbiotic, the exact microbial composition must be characterized before inactivation processes: indeed, preparations obtained from undefined microorganisms do not fit with the postbiotic definition. On the contrary, metabolites generated during intestinal bacteria fermentation, i.e., secreted proteins and short-chain fatty acids (SCFAs), or molecules representing structural fragments of these bacteria, such as exopolysaccharides (EPS) and lipoteichoic acid (LTA), could be considered as postbiotics, as long as they are in the presence of the inactivated microbial cell and/or its cell components, too [9,10].

Thus, while probiotics are living nonpathogenic microorganisms (such as *Saccharomyces boulardii* yeast or *Lactobacillus* and *Bifidobacterium* species) able to provide health benefits if assumed in adequate amounts [11] and prebiotics are non-digestible substances (i.e., dietary fibers) that promote probiotics’ growth, contribute to gut health and exert immunomodulatory effects [12], postbiotics refer to bioactive compounds produced by microorganisms during their growth, including the inactivated form of the microbial cells—that does not necessarily mean that they need to be qualified as a probiotic (while living) to be accepted as a postbiotic. Currently, inanimate strains of genera from the Lactobacillaceae family and strains of the genus *Bifidobacterium* represent the bulk of known postbiotics [13].

## 2. Classification of Postbiotics

As mentioned above, it is possible to identify two main categories of postbiotics: (i) metabolites, derived from intestinal bacterial fermentation, and (ii) structural components, as intrinsic constituents of the microbial cell [14]. Despite SCFAs, vitamins and peptides represent the best characterized postbiotics, and a large variety of bioactive molecules fitting the definition of postbiotics have been identified (Table 2).

### 2.1. Short-Chain Fatty Acids (SCFAs)

Short-chain fatty acids (SCFAs) are a subset of fatty acids derived from the fermentation of partially and non-digestible polysaccharides (dietary fibers and resistant starches) carried by the enzymatic activity of specific taxa belonging to Bacteroidetes and Firmicutes species [18]. The carbon chain of SCFAs, for definition, is composed of less than six carbons, acetate (C2), propionate (C3) and butyrate (C4) being the most represented ones [19]. Indeed, these three SCFAs contribute to 80% of the total SCFA pool in the human body (~60% acetate, ~20% propionate and butyrate) [20].

The composition of SCFAs’ pool fluctuates throughout life, depending both on diet variety and gut microbiota composition [21]: high fiber–low fat diets expand the SCFAs’ pool and increase the fecal and blood SCFA content (fSCFAs, bSCFAs); a diet with low fiber content does the opposite [14,22]. However, obesity is related to increased levels of total fSCFAs, too [23,24], which are reduced following an anti-obesity treatment [25]. Also, changes in the intestinal microbiota, as mentioned above, have a direct impact of SCFAs’ pool heterogeneity: early stages of life (0–3 years) are characterized by high levels of acetate, mainly produced by Bifidobacteria (in particular *Bifidobacterium breve* and *Bifidobacterium bifidum*), due to human milk oligosaccharide (HMO) consumption in breastfeeding [26,27]; in adulthood, increased levels of propionate and butyrate positively correlate with an increase in Firmicutes (e.g., *Lactobacillaceae*, *Ruminococcaceae*, *Lachnospiraceae)* [28,29,30] and Bacteroidetes (e.g *Bacteroides fragilis*, *Bacteroides thetaiotaomicron*, *Bacteroides caccae*, *Bacteroides ovatus*) [31,32,33], which are reduced at the elderly stage due to a decrease in microbial diversity of commensal taxa accompanied by an increased abundance of pathobionts such as *Enterobacteriaceae* and *Streptococcus* spp. [34,35,36].

Thanks to their ability to bind several cell-surface G protein-coupled receptors (GPCRs) [37], SCFAs are responsible for the development of a tolerogenic immune system by (i) promoting regulatory T (Treg) cells and quiescent dendritic cell (DC) phenotypes; (ii) enhancing epithelial barrier function; (iii) stimulating mucus secretion by intestinal goblet cells; (iv) damping inflammatory response and suppressing autoimmune reactions [38,39,40,41] (Table 3).

Moreover, SCFAs have been proved responsible for increasing insulin sensitivity by stimulating the production of insulin-sensitizing hormones like glucagon-like peptide 1 (GLP-1) and by reducing inflammation in adipose tissue [59,60]; they promote hepatic and muscular FA oxidation and adipose tissue lipolysis while inhibiting hepatic FA synthesis, contributing to lipid metabolism homeostasis [61,62]. Last, SCFAs can be converted into acetyl-CoA, thus entering the Krebs Cycle and participating in energy production [59]. An overview of the main functions of SCFAs is shown in Figure 1.

### 2.2. Lactic Acid

Lactic acid (LA) is a carboxylic acid generated from pyruvate, the end product of glycolysis, by the activity of the lactate dehydrogenase (LDH) enzyme [63]. In hypoxia or anaerobic conditions, when oxygen is not available as the final acceptor of electrons, pyruvate conversion to lactate allows the regeneration of nicotinamide adenine nucleotide (NAD^+^), essential for sustained glycolysis [64].

Lactic acid is also produced via fermentation of multiple carbohydrate sources (e.g., glucose, cellulose, xylose, maltose, lactose and others) by different Gram-positive, catalase-negative, non-spore-forming bacteria with amylase activity referred to as lactic acid bacteria (LAB) [65,66]. In addition to LA, LAB generate other growth inhibitions molecules such as bacteriocins, antifungal peptides and hydrogen peroxide (H_2_O_2_), preventing the proliferation of putative pathogen microorganisms [67,68]. *Lactobacillus* strains are the most well-known LA producers, even if *Lactococcus*, *Streptococcus*, *Pediococcus* and *Enterococcus* are important, too [69]. Acting like an acidifying metabolite, LA is essential for the maintenance of a healthy vaginal environment, preventing the colonization or growth of pathogens responsible for common vaginal infections, such as *Gardrenella vaginitis* and *Candida albicans* [70].

### 2.3. Bacteriocins

Bacteriocins are a heterogeneous group of antimicrobial peptides (AMPs) naturally secreted by both Gram-positive and Gram-negative species belonging primarily to the genera *Bifidobacterium* and *Lactobacillus* [71,72]. Bacteriocins are synthesized with the purpose of killing other bacteria, particularly pathogenic ones such as *Staphylococcus aureus*, *Pseudomonas aeruginosa*, *Salmonella typhi*, *Listeria monocytogenes* and *Clostridium botulinum*, thus participating in microbial competition among prokaryotes [73,74]. Bacteriocins are very diverse in terms of length, molecular weight, genetic origins, immunity mechanisms, biochemical and/or structural features and may act via multiple modes of actions, such as pore formation on target cell membrane, inhibition of cell wall synthesis as well as nucleic acid degradation through DNase and RNase activity [65,72,75]. Differently from antibiotics, bacteriocins offer more benefits as they are natural bioactive peptides with no side effects and represent a possible solution to Multiple Drug Resistance (MDR) disease-causing bacteria, more than having multiple additional positive effects on human health [76,77]. A list of the best known bacteriocins is reported in Table 4.

### 2.4. Secondary Bile Acids

Secondary bile acids, specifically deoxycholic acid (DCA) and lithocholic acid (LCA), are steroidal bacterial metabolites produced in the colon derived from the conversion of primary bile acids cholic acid (CA) and chenodeoxycholic acid (CDCA) carried by specific bile salt hydrolase (BSH)-expressing microorganisms [89,90] (Figure 2). Despite the most well-known purpose of bile acids being to facilitate dietary lipid emulsion and absorption, recent studies demonstrated that bile acids have several biochemical and immunological effects by acting as ligands of both membrane and nuclear receptors referred to as Bile Acid Receptors (BARs) [91,92,93]. This function will be discussed in more detail later.

### 2.5. Bacterial Cell Wall Components (CWCs)

Recent studies include bacterial cell wall components (CWCs) in the postbiotic class: among these, exopolysaccharides (EPSs), peptidoglycan (PG) and lipoteichoic acid (LTA) are the most relevant ones [95].

EPSs are high-molecular-weight carbohydrate biopolymers synthesized by microorganisms that can be classified into (i) capsular polysaccharides, closely associated with the cell surface, and (ii) free slime polysaccharides, loosely attached or totally secreted into the extracellular environment [96]. Species belonging to both Gram^+^ and Gram^−^ bacteria produce EPSs, such as *Acetobacter*, *Gluconobacter*, *Pseudomonas*, *Enterobacter*, *Klebsiella*, *Bacillus*, *Streptococcus* and *Clostridia* genera [97]. Recent studies demonstrated how EPSs from *Lactobacillus brevis* could lower intestinal pH, upregulate SCFA production, especially propionate and butyrate via improving intestinal microbiota [98], while EPSs from *Bifidobacterium longum* managed to alleviate DSS-induced intestinal inflammation in mouse model modulating macrophage polarization toward the anti-inflammatory M2-type [99].

PG is a three-dimensional polymer representing the primary component of the Gram^+^ and Gram^−^ bacteria cell wall, responsible for cell shape maintenance and resistance to both extracellular environmental insults and intracellular osmotic pressure (or turgor) caused by cytosolic content [100,101,102]. PG administration in vitro results in the inhibition of pro-inflammatory cytokine release such as interleukin 6 (IL-6), IL-8, IL-1β and tumor necrosis factor α (TNFα) [103] while upregulating anti-inflammatory genes, including IL-10 and transforming growth factor β (TGF-β) [104]. In addition, PG promotes collagen synthesis, fibroblast proliferation and angiogenesis, thus promoting wound healing and tissue regeneration [105,106].

LTA, instead, is a surface-associated adhesion amphiphilic molecule found exclusively in Gram^+^ bacteria that works to maintain ion homeostasis, resist osmotic stress and regulate autolytic activity [107,108]. Studies showed that LTA from *Lactobacillus plantarum* exerts anti-inflammatory activity both in vitro and in vivo, by reducing Toll-like receptor 2 (TLR2) and subsequent nuclear factor-κB (NF-κB) activation in human intestinal epithelial cells [109,110], lowering the Colitis Disease Activity Index (CDAI) as well as TNFα levels in LTA-treated mice compared to untreated ones [110].

### 2.6. Plasmalogens (Pls)

Plasmalogens (Pls) are a unique class of membrane glycerophospholipids characterized by the presence of a fatty alcohol and several polyunsaturated fatty acids bound to the glycerol backbone [111]. In bacteria, major evidence supports the theory that Pls play an important role in exosome fission [112]. In the human body, Pls are mainly expressed in the heart, retina and innate immune cells and represent the main component (up to 80%) of neural tissue [113,114]. Pls account for around 20% of the total human phospholipids and play important roles in cell homeostasis, cell signaling and neural transmission [115,116]. Daily oral administration of Pls as postbiotics seems to (i) regulate adipogenesis [117], (ii) have anti-inflammatory and antioxidant activities [118] as well as (iii) improve cognitive function in patients with mild Alzheimer’s disease [119].

### 2.7. Intestinal Bacteria-Derived Vitamins

Vitamins are organic micronutrients defined as essential constituents of the diet not endogenously synthesized by humans or not synthesized in an adequate amount to support human health [120,121]. Based on their biochemical composition, vitamins can be classified as water-soluble (B1 B2, B3, B5, B6, B7, B9, B12 and C) or fat-soluble (A, D, E and K) [122,123,124]. However, vitamins are not only introduced via exogenous sources: several gut bacteria, belonging predominantly to *Bacteroides*, *Bifidobacterium* and *Enterococcus* genera, contribute to vitamin synthesis with particular reference to thiamine (B1), riboflavin (B2), pantothenic acid (B5), biotin (B7), folate (B9), cobalamin (B12) and vitamin K [125,126]. Given the multiple metabolic functions of both hydro- and lipo-soluble vitamins in human health such as (i) immune-modulation, (ii) bone health maintenance, (iii) calcium balance, (iv) retinal health and sight protection, (v) antioxidant activity and (vi) blood clotting regulation, adequate amounts of vitamin uptake should be guaranteed to avoid syndromes and/or diseases derived from vitamin deficiencies [121,127,128,129].

### 2.8. Tryptophan Metabolites

Tryptophan is a widely investigated amino acid, essential for body health and homeostasis: it cannot be synthesized de novo by human cells but it must be supplied through the diet via bread, milk, chocolate, tuna fish and other foods rich in such amino acid [130,131]. Tryptophan is required for a normal body’s growth and development, being the in vivo precursor of several bioactive compounds such as nicotinamide (B6), serotonin, melatonin, tryptamine, kynurenine and others [132,133,134] as well as affecting metabolism of neurotransmitters and CNS compounds such as dopamine, norepinephrine and beta-endorphin [135,136,137]. Tryptophan is also converted into indoles via the action of the tryptophanase (TnaA) enzyme, expressed in various both Gram^+^ and Gram^−^ bacteria species including *Escherichia coli*, *Lactobacillus* spp., *Clostridium* spp. and *Bacteroides* spp. [138,139,140].

Tryptophan derivatives exert multiple beneficial effects: (i) they regulate neurotransmitters levels, thus having positive influence on the recognition of positive emotions [141,142]; (ii) they regulate both innate and adaptive immunity towards an antimicrobial, anti-inflammatory and tumor surveillance phenotype via kynurenine [143]; (iii) they enhance the function of the intestinal epithelial barrier via indoles [144] and inhibit LPS-induced pro-inflammatory interleukin expression through Aryl hydrocarbon receptor (AhR) signaling [145]; (iv) they improve insulin resistance [146] and lipid metabolism, reducing liver steatosis and inflammation, thus alleviating metabolic dysfunction-associated steatotic liver disease (MASLD) [147,148,149].

### 2.9. Conjugated Linoleic Acids (CLAs)

Conjugated linoleic acids (CLAs) refer to a pool of cis or trans isomers of the polyunsaturated omega-6 essential fatty acid linoleic acid, cis-9, cis-12 and octadecadienoic acid being the most represented ones (almost 95% of all linoleic acid isomers) [150]. CLAs derive from the biohydrogenation of linoleic acid carried by bacteria that express linoleic acid isomerase, such as *Bifidobacterium breve*, *Bifidobacterium infantis*, *Bifidobacterium adolescentis*, *Lactobacillus reuteri*, *Roseburia* spp. and others [151,152].

When administered as postbiotics, CLAs show multiple beneficial effects on human health, including: (i) anti-breast cancer properties [153]; (ii) body fat reduction via increased lipolysis and decreased FA accumulation in adipose tissue [154]; (iii) atherosclerosis inhibition [155]; (iv) improved immune system functions and reduced inflammation [156]; (v) osteoporosis prevention [157] and many others [158].

### 2.10. Polyamines

Polyamines are organic polycationic alkylamines synthesized from L-ornithine and/or arginine or by amino acid decarboxylation that play important roles in a huge variety of biological functions in all organisms, from cell metabolism to apoptosis and cell differentiation [159,160]. Among all, putrescine (PUT), spermine (SPE), spermidine (SPD) and cadaverine (CAD) are the most important ones and recent studies demonstrated their role as NLRP6 inflammasome inhibitors [161,162].

Interestingly, the human gut microbiota, with bacteria belonging to *Bifidobacterium*, *Clostridium*, *Enterococcus*, *Lactobacillus* and *Enterobacter* expressing arginine decarboxylase and/or ornithine decarboxylase, is a major contributor to the total polyamine pool in vivo [163,164].

Polyamines are essential for intestinal epithelial renewal and barrier integrity and homeostasis, acting through both transcriptional and posttranscriptional control of expression of multiple genes involved in intestinal epithelial cell (IEC) proliferation, migration and cell-to-cell interactions [165]. PAs are also fundamental for a proper immune system development, since PA depletion causes abnormal differentiation of cytolytic T lymphocytes and defective immunoglobulin-producing B cells [166,167,168]. In addition, SPE is able to reverse B cell senescence [169].

Despite that, elevated levels of polyamine can inhibit immune cell activity and have been associated with tumorigenesis, in particular with breast, colon, prostate and skin cancers [170,171].

### 2.11. Phenolic Compounds

Phenolic compounds (or polyphenols) are a heterogeneous group of natural bioactive molecules defined as secondary metabolites mainly found in plant tissues and generated during plant metabolism that play a pivotal role in protecting from UV radiations and pathogen aggression [172]. Polyphenols are largely found in fruits, vegetables and cereals, thus representing an important component of our diet [173].

After dietary ingestion, polyphenols are metabolized by the human gut microbiota (e.g., *Eubacterium ramulus*, *Lactobacillus* spp. and *Gordonibacter urolothinfaciens*) via multiple biotransformation processes such as esterification, glycosylation, hydrolysis and acylation [174]. Urolithins, derived from the microbiota transformation of ellagitannins (ETs) and ellagic acid (EA), represent one of the most common and important polyphenol-derived group of metabolites which have drawn the attention of the scientific community for the last few years for their pleiotropic health beneficial effects in preventing several conditions such as cardiovascular diseases (CVDs), diabetes, aging, asthma and infectious diseases thanks to their well-known antioxidant, anti-inflammatory, neuroprotective and cardioprotective effects [175,176,177].

### 2.12. Hydrogen Peroxide (H_2_O_2_)

Hydrogen peroxide (H_2_O_2_) is an endogenous reactive oxygen species (ROS) which naturally occurs as a byproduct of cellular respiration [178]. H_2_O_2_ contributes to oxidative stress both directly, acting as a molecular oxidant (e.g., peroxidation of membrane lipids which leads to membrane integrity disruption), and indirectly, through free radical generation, which penetrates cell membranes and reacts with intracellular molecules [179]. Moreover, in concentrations from 1% to 6%, H_2_O_2_ has antimicrobial properties [180].

*Lactobacillaceae* and their H_2_O_2_ production represent one of the most important mechanisms of colonization resistance against pathogen microbes, thus making it possible to be considered an interkingdom antivirulence strategy [181].

However, given its oxidizing activity, H_2_O_2_ is responsible for single- and double-strand DNA breaks and it seems to play a pivotal role in mutagenesis and tumorigenesis of thyroid cells, specifically when proper antioxidant defenses are lacking [182]. Moreover, in vitro studies suggest that exposure of cortical neural cells to H_2_O_2_ is toxic, being responsible for increased intracellular free calcium concentration and apoptotic cell death within 3 h [183]. Lastly, elevated levels of H_2_O_2_ cause a decrease in gap junction (GJ) resistance and well as a reduction in intracellular pH, leading to acidosis [184].

### 2.13. Organic Acids

Organic acids are a heterogenous class of low-molecular-weight (LMW) compounds containing at least one carboxylic acid group which are intermediate products of several cellular catabolic pathways including glycolysis, tricarboxylic acid (TCA) cycle and FA oxidation [185,186].

Similarly to H_2_O_2_, organic acids such as formic acid (FA), mainly produced by *Lactobacillus* spp. [187], are suitable as antibacterial agents by acting as inhibitory metabolites through colonization resistance mechanisms by preventing pathogens’ enzymes from working properly [188,189].

### 2.14. Glutathione (GSH)

γ-L-glutamyl-L-cysteinyl-glycine, known as glutathione (GSH), is one of the most important LMW antioxidant compounds produced by the cell. It is obtained by the sequential addition of cysteine and glycine to a glutamate molecule and its potential is majorly due to the sulfhydryl group (-SH) of the cysteine residue which is involved in reduction and conjugation reactions, making GSH essential for peroxide removal and xenobiotic metabolism [190,191].

Recent studies demonstrated how *Lactobacillus salivarius* can enhance GSH de novo synthesis, which in turn inhibits mitochondrial biogenesis in osteoclasts (OCs), thus representing an interesting approach for the treatment and prevention of osteoporosis [192]. Due to its antioxidant activity, GSH supplementation may be used as an anti-inflammatory and immunomodulatory compound [193,194,195].

### 2.15. Microbial Enzymes

Enzymes are proteins required to accelerate metabolic processes by decreasing activation energy for a chemical reaction to occur. This allows to speed up reaction rates; reactions that, otherwise, would not be time-compatible with physiological biological timelines [196].

Microbial enzymes are particularly interesting due to economic feasibility, high yields, rapid bacteria growth rates and inexpensive culture media as well as greater catalytic activity [197]. Microbial enzymes mediate several metabolic, physiological and regulatory processes and are able to resist to unusual temperatures and pH conditions, making them attractive not only for medical but also for industrial applications [198]. Some of the most common enzymes used as postbiotics are proteases and lipases.

Proteases, mainly synthesized by *Lactobacillus* spp. bacteria, not only participate in protein-rich food digestion: they have been shown to be responsible for the generation of bioactive peptides (2–20 amino acids) with immunomodulatory and anticancer activities [199].

Lipases, produced by bacterial *Bacillus* spp., *Alcaligens* spp., *Pseudomonas* spp. and fungi *Penicillium* spp. and *Aspergillus* spp., may act as postbiotics by exerting multiple effects via antioxidant influence, antimicrobial and lipolytic action [200]. Lipases may help in reducing inflammation via lipid metabolism modulation in conditions such as obesity or metabolic syndrome [201,202].

## 3. Secondary Bile Acids as Postbiotics

Bile acids are amphipathic molecules derived by cholesterol conversion via a chain of enzymatic reactions within hepatocytes: as a result, primary bile acids chenodeoxycholic acid (CDCA) and cholic acid (CA) are obtained and then conjugated with glycine (G) or taurine (T) residues to give rise to their respective bile salts which are finally secreted into the bile [203,204,205].

Other than being involved in dietary lipid emulsion and adsorption, once they have reached the small intestine, primary bile acids are processed and metabolized into secondary bile acids by intestinal microbial enzymatic activity [206]: such biotransformations are mainly represented by deamination (or deconjugation), carried by the Bile Salt Hydrolase (BSH) abundantly expressed by *Lactobacillus*, *Bifidobacterium*, *Enterococcus* and *Clostridium* species [207], and epimerization, carried out by hydroxysteroid dehydrogenases (HSDHs) [208]. While LCA, DCA and ursodeoxycholic acid (UDCA) are the best characterized secondary bile acids, actually, more than 692 novel bile acids [209], including over 200 microbiota-derived secondary bile acids [210,211] (MDBA), have been identified [209,212].

Thanks to their ability to bind a heterogeneous family of both membrane and nuclear receptors referred to as bile-acid-regulated receptors (BARRs) [213], which are ubiquitously expressed by different cell types of the human body such as enterocytes, hepatocytes, neurons, adipocytes and immune cells [214], secondary bile acids represent the most abundant families of chemical metabolites able to mediate mutual interactions between the intestinal microbiota and the host, regulating immune system, glucose and energy metabolism [215,216,217]. The two best characterized BARRs are FXR [218] and GPBAR1 (also known as TGR5) [219]. FXR functions as a bile acid sensor [94,220], regulates bile acid synthesis and homeostasis, and is mainly activated by primary bile acids [221]. In contrast, GPBAR1 [219] regulates energy expenditure [222] and glucose metabolism and is preferentially activated by secondary bile acids. In addition to these metabolic effects, both FXR and GPBAR1 exert immunoregulatory effects [203,213,223] in the liver, intestine [224] and cardiovascular system [225] (Figure 3).

### 3.1. Secondary Bile Acids and Immunity

Due to the expression of different BARRs by both innate and adaptive immunity cells, secondary bile acids are responsible for the development of a tolerogenic immune system. Specifically, different secondary bile acids have different affinities for various receptors, each one responsible for the activation of specific pathways (Table 5).

DCA and LCA are the main physiological ligands of GPBAR1 in humans [219]. GPBAR1 is expressed by monocytes, macrophages, DCs and natural killer T (NKT) cells [267]. GPBAR1 activation upon bile acids binding on these cells promotes the development of a tolerogenic phenotype in the immune system via different mechanisms: (i) acting as a negative regulator of the pro-inflammatory NF-κB pathway by inhibiting IκBα phosphorylation and p65 nuclear translocation [268,269]; (ii) inducing CREB phosphorylation, responsible for NF-κB-responsive element repression [270]; (iii) inhibiting NLR family pyrin domain containing 3 (NLRP3) inflammasome, thus preventing the secretion of pro-inflammatory mediators like IL-6, IL-1 β, TNF-α [232]; (iv) stimulating, in monocytes and macrophages, the secretion of the IL-10 anti-inflammatory cytokine [226].

LCA and its derivative 3-keto-LCA are ligands for the Pregnane-X-Receptor (PXR), expressed by monocytes, macrophages, CD4^+^, CD8^+^ and B cells [271]. Similarly to GPBAR1, PXR activation inhibits NF-κB and NLRP3 inflammasome assembly [272]. Also, LCA binds and activates the Constitutive Androstane Receptor (CAR), predominantly expressed by T cells, inducing effector T cell reprogramming, IL-10 secretion increase and Treg cell pool expansion [273]. In addition, Vitamin D Receptor (VDR) is activated by LCA and its derivatives, too, reducing pro-inflammatory cytokine expression in monocytes, macrophages and Kupffer cells (KCs) while promoting Tregs expansion by increasing FOXP3 expression [203,252].

Simultaneously, LCA, DCA and their derivatives (3-oxo-LCA, iso-allo-LCA and iso-allo-DCA) contribute to the development of a tolerogenic reprogramming of the immune system by acting as inverse agonists for the Retinoid Orphan Receptor gamma T (RORγt), a typical transcription factor responsible for Th17 differentiation [274,275], thus inhibiting Th17 differentiation while promoting Treg expansion via FOXP3 expression [276,277].

### 3.2. Secondary Bile Acids and Glucose Metabolism

Intestinal FXR and GPBAR1 expression is essential for a proper regulation of glucose metabolism via the secretion of two enterokines, the Fibroblast Growth Factor (FGF)-19 [278] and Glucagon-Like Peptide (GLP)-1 [203,279]. In post-prandial conditions, primary and secondary bile acids form in the gastrointestinal tract bind and activate intestinal FXR [280,281,282] and GPBAR1 [283], promoting the release of FGF-19 from ileal enterocytes and GLP-1 from ileal and colonic L cells, respectively. Binding to its FGF receptor (FGFR) on hepatocytes, FGF-19 acts as a CYP7A1 repressor, thus contributing to the feed-back inhibition of bile acid synthesis [284,285]. Various FXR agonists are currently developed for clinical applications [220,286,287,288], although animal studies seem to suggest that FXR antagonists [289,290,291] might also have a potential therapeutic utility.

On the other hand, GLP-1 stimulates insulin secretion from pancreatic ϐ-cells promoting glucose uptake, delays in gastric emptying and appetite suppression [292]. Moreover, recent studies showed that pancreatic ϐ-cells express, themselves, both FXR and GPBAR1, thus making it possible for primary and secondary bile acids to directly induce insulin transcription and secretion [293,294].

### 3.3. Secondary Bile Acids as Exercise Mimetic and Longevity-Associated Molecules

The composition of intestinal microbiota is influenced by physical activity and dietary intake. In general, physical activity reduces the beta-diversity of gut microbiota composition and postbiotic production, particularly SCFAs. However, in addition to SCFAs, secondary bile acids are increasingly recognized as potent modulators of energy expenditure and metabolism.

GPBAR1 is robustly expressed in thermogenic competent tissues including striated muscle and white (WAT) and brown (BAT) adipose tissues [222,295]. In these tissues, GPBAR1 agonism by DCA and LCA promotes a cAMP-dependent expression of the type 2 iodothyronine deiodinase (DIO2), a thyroid hormone activating enzyme responsible for tetraiodothyronine (T4) conversion into active tri-iodothyronine (T3) [222]. T3 binds and activates the thyroid hormone receptor (THR) that acts as a transcription factor for various genes, increasing both energy expenditure via thermogenesis and basal metabolic rate. A key transcription factor involved in this effect is the Uncoupling Protein 1 (UCP1) [296].

UCP1 plays a crucial role as an *exercise mimetic* by mimicking some of the metabolic and thermogenic benefits of physical activity, particularly through its role in energy expenditure and metabolic regulation [296]. UCP1 is a mitochondrial protein primarily expressed in the BAT and beige fat, where it dissipates the proton gradient to generate heat instead of adenosine triphosphate (ATP). This process, known as non-shivering thermogenesis, mimics the metabolic effects of exercise by increasing caloric burn and lipid oxidation. UCP1 activation improves whole-body glucose metabolism and protects against insulin resistance, resembling the effects of endurance training. In muscle cells, UCP1 enhances mitochondrial biogenesis and functions in skeletal muscle; targeting UCP1 pharmacologically (e.g., with β3-adrenergic agonists or cold exposure) is an area of interest for combating obesity and metabolic diseases in individuals who are unable to exercise. The two main secondary bile acids DCA and LCA and their isoforms exert metabolic effects that mimic exercise by activating GPBAR1 and FXR signaling (including UCP1), promoting mitochondrial function, enhancing glucose metabolism and reducing inflammation. While they do not replace physical activity, pharmacological modulation of secondary bile acid signaling could serve as a therapeutic strategy for metabolic disorders, obesity and age-related metabolic decline, especially for individuals unable to exercise.

This view is further supported by the observation that secondary bile acids, specifically LCA, might be the mediator of beneficial effects exerted by calorie restriction on body weight and life span [16]. A recent study has shown that LCA accumulates in muscles and BAT and WAT in response to calorie restriction, while its administration promotes body weight reduction via activated protein kinase (AMPK)-dependent pathways [297]. Recent evidence suggests that LCA is able to recapitulate, at least in animal models, all the beneficial effects of calorie restriction [16,298], a dietary intervention that can promote overall health and lifespan extension through the reduction of inflammation and reactive oxygen species (ROS) production, generally altered in age-related metabolic disorders and immune mediated diseases [299,300,301]. Qu et al. demonstrated that LCA administration activates AMPK in muscle cells, enhancing muscle regeneration and strength in old mice as well as inducing life extension in *Caenorhabditis elegans* and *Drosophila melanogaster*, all effects abrogated after knocking-down AMPK [16]. Additionally, LCA enhances mitochondrial respiration and reduces reactive oxygen species (ROS), thereby improving cellular energy metabolism and expression of mitochondrial unfolded protein response (UPRmt) in *C. elegans*. It has also been shown that LCA might promote a sirtuin-dependent activation of AMPK. LCA induces sirtuin 3, a mitochondrial deacetylase that enhances energy metabolism and oxidative stress resistance. It is important to consider that while LCA could extend the lifespan of yeast and *Caenorhabditis elegans*, evidence in higher organisms such as mice and humans is still limited.

In addition to sirtuin 3, Qu et al. have shown that LCA activates sirtuin 1 [298] and identified TUB-like protein 3 (TULP3) as receptors for LCA. Specifically, LCA binding to TULP3 induces the allosteric activation of sirtuins, which subsequently deacetylates the V1E1 subunit of v-ATPase at residues K52, K99 and K191, promoting a robust activation of AMPK and muscle function in aged mice.

Partially in accordance with these results, a study carried out in 2021 in Japanese centenarians detected elevated fecal levels of LCA and its metabolites. Given the potent antimicrobial effects of iso-LCA, it has been proposed that this could contribute to centenarians’ long-lasting health [208]. Nevertheless, it is important to remember that the high iso-allo-LCA hydrophobicity has been associated with an increased risk for age-related cognitive impairment [302].

While these studies suggest that secondary bile acids have beneficial effects on metabolism, and the majority of reports published in the last decade envision LCA and DCA and their receptors as potential targets to treat metabolic and inflammatory disorders, there is older literature that raises concerns over the potential harmful effects of these relative hydrophobic bile acids [303]. Bile acids have been reported to promote inflammation, rather than reducing the activation of inflammatory cells including direct activation of inflammasome. However, some of these effects are obtained at relatively high concentrations, approximatively 100 µM or higher, while the EC50 for the activation of nuclear and G-protein coupled receptors are generally in nanomolar or low micromolar ranges.

In addition to these beneficial effects, early evidence has also raised some concerns over the possibility that secondary bile acids might be detrimental for the human intestine and might exert a role in the development of intestinal injury and cancers. High concentrations of bile acids promote cell damage, oxidative stress, ROS production and DNA damage [304], and have been considered as putative etiologic agents in the development of gastrointestinal cancers, including esophageal, gastric, liver, bile duct, pancreatic and colorectal cancers [305]. Consistent with this view, both prospective and retrospective studies in humans associate high circulating blood levels of secondary bile acids with increased risks of colon cancer [306]. However, it should be noted that more recent studies carried out in IBD patients have shown that IBD development associates with intestinal dysbiosis and reduced excretion of secondary bile acids [223,307]. Further on, secondary bile acids restrain intestinal inflammation in IBD patients and models, by GPBAR1, VDR and RORγT-dependent mechanisms [274].

## 4. Conclusions

The intestinal microbiota is a source of an extraordinary variety of bile acids. Gut bacteria transform primary bile acids into secondary bile acids, which then act as metabolic regulators through GPBAR1, PXR and RORγT and other canonical receptors such as VDR [224]. A balanced microbiota promotes healthy metabolism, while intestinal dysbiosis might participate in the development of several human disorders including obesity, diabetes and systemic inflammation. By modifying the intestinal microbiota or bile acid pathways, it might be possible to prevent and treat metabolic diseases. Secondary bile acids modulate inflammation, instructing the host immune system and regulating glucose and energy metabolism, making it possible to define them as one of the most interesting classes of postbiotics. Secondary bile acids are emerging as potent regulators of metabolism, inflammation, and immunity. By modifying the intestinal microbiota or designing bile acid-based drugs, it will be possible to design novel therapies for metabolic disorders, inflammatory diseases. These bile-acid–microbiota interactions’ growth could lead to transformative treatments for a wide range of human diseases.

## Figures and Tables

**Figure 1 cells-14-00595-f001:**
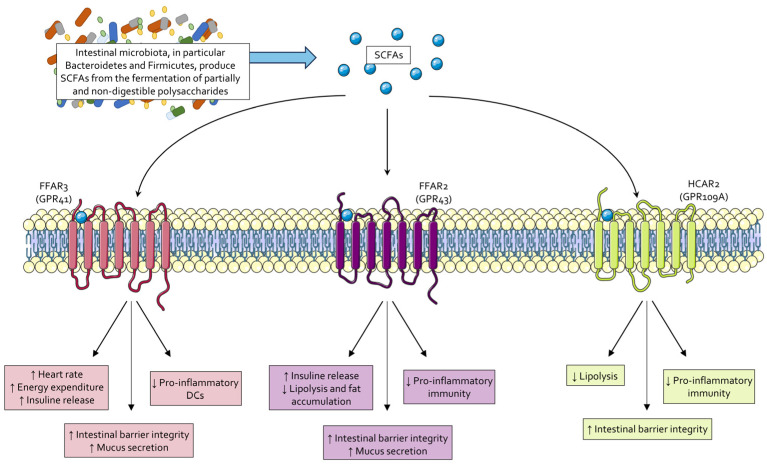
Effects of SCFAs. Intestinal microbiota bacteria, particularly species within the Bacteroidetes and Firmicutes phyla, produce short-chain fatty acids (SCFAs) through the fermentation of partially and non-digestible polysaccharides. SCFAs function as signaling molecules by activating three membrane receptors: FFAR3 (also known as GPR41), FFAR2 (also known as GPR43), and HCAR2 (also known as GPR109A). The activation of these receptors elicits anti-inflammatory effects, preserves intestinal barrier integrity, and reduces fat accumulation by enhancing insulin secretion and energy expenditure.

**Figure 2 cells-14-00595-f002:**
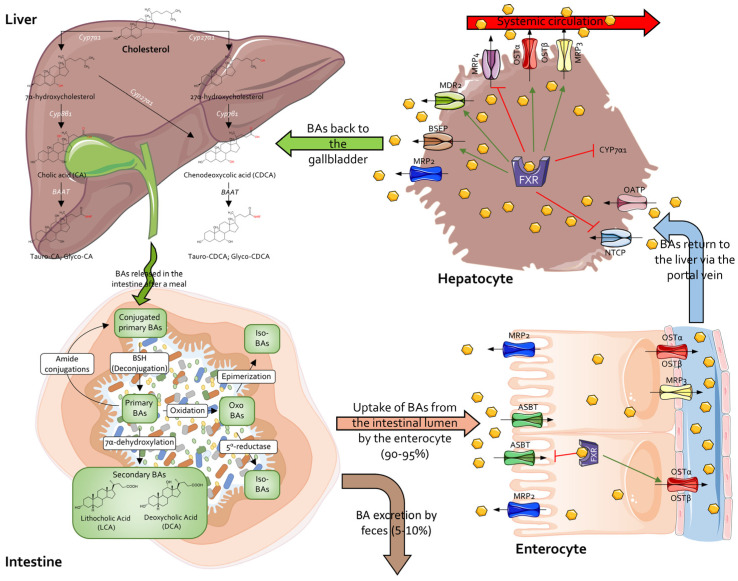
Bile acid biosynthesis and enterohepatic recirculation. Primary bile acids—cholic acid (CA) and chenodeoxycholic acid (CDCA)—along with their taurine and glycine conjugates, are synthesized in the liver from cholesterol via two distinct metabolic routes. The classical (neutral) pathway is initiated by CYP7A1, whereas the alternative (acidic) pathway begins with CYP27A1. Following a meal, these bile acids are secreted into the small intestine through the common bile duct, where they facilitate lipid emulsification and absorption. In the gut, microbial enzymes further transform bile acids, yielding secondary bile acids such as deoxycholic acid (DCA) and lithocholic acid (LCA) along with their derivatives. In the ileum, bile acids are absorbed by enterocytes via the ASBT transporter; this uptake activates the farnesoid X receptor (FXR), which in turn downregulates ASBT to restrict further reabsorption. Subsequently, bile acids are conveyed from enterocytes to the portal vein through MRP3 and OSTα/β transporters, returning to the liver where hepatocytes reabsorb them via OATPs and NTCP. Elevated intracellular bile acid concentrations in hepatocytes activate the FXR/SHP signaling axis, thereby suppressing further bile acid synthesis through inhibition of CYP7A1. Moreover, hepatocytes export bile acids into the systemic circulation via MRP4 [94], OSTα/β and MRP3, while also exporting them back into bile ducts through MDR2, MRP2 and BSEP. Abbreviations: CYP7A1, cytochrome P450 7A1; CYP27A1, cytochrome P450 27A1; CYP8B1, cytochrome P450 8B1; CA, cholic acid; CDCA, chenodeoxycholic acid; BAAT, bile acid-CoA:amino acid N-acyltransferase; BA, bile acid; BSH, bile salt hydrolases; HSDH, 7α-hydroxysteroid dehydrogenase; ASBT, apical sodium-dependent bile acid transporter; MRP2, multidrug resistance-associated protein 2; FXR, farnesoid X receptor; MRP3, multidrug resistance-associated protein 3; NTCP, sodium/taurocholate co-transporting polypeptide; OATPs, organic anion-transporting polypeptides; MDR2, multidrug resistance protein 2; BSEP, bile salt export pump; MRP4, multidrug resistance-associated protein 4.

**Figure 3 cells-14-00595-f003:**
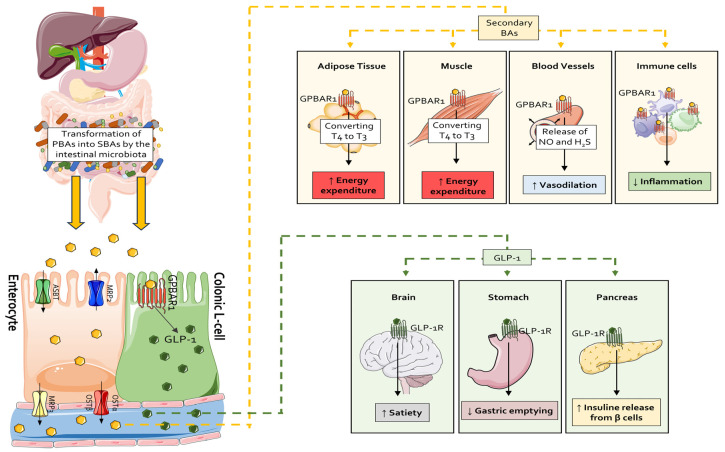
Systemic effects of secondary bile acids. Within the gastrointestinal tract, the resident microbiota converts primary bile acids into their secondary counterparts. Nearly 90% of these bile acids are reabsorbed in the ileum by enterocyte and returned to the liver through the portal circulation, a process facilitated by OATP and NTCP transporters. In the colon, secondary bile acids engage GPBAR1 receptors on enteroendocrine L-cells, thereby triggering GLP-1 secretion. GLP-1 subsequently enhances satiety, decelerates gastric emptying and augments insulin release via activation of GLP-1R on neuronal networks, gastric tissues and pancreatic β-cells. Moreover, systemic activation of GPBAR1 in adipose tissue, skeletal muscle, blood vessels and immune cells by secondary bile acids promotes, respectively, thermogenic activity through T4-to-T3 conversion and vasodilation and confers anti-inflammatory effects.

**Table 1 cells-14-00595-t001:** Definitions of postbiotics before the 2019 ISAPP consensus.

Definition of Postbiotics	Microbial Cells Included	Reference
Any factor resulting from the metabolic activity of a probiotic or any released molecule capable of conferring beneficial effects to the host in a direct or indirect way	No	[2]
Soluble factors (products or metabolic byproducts), secreted by live bacteria, or released after bacterial lysis, such as enzymes, peptides, teichoic acids, peptidoglycan-derived muropeptides, polysaccharides, cell surface proteins and organic acids	No	[3]
Compounds produced by microorganisms, released from food components or microbial constituents, including non-viable cells that, when administered in adequate amounts, promote health and well-being	Yes	[4]
Non-viable metabolites produced by probiotics that exert biological effects on the hosts	No	[5]
Non-viable bacterial products or metabolic byproducts from probiotic microorganisms that have positive effects on the host or microbiota	Yes	[6]
Functional bioactive compounds, generated in a matrix during fermentation, which may be used to promote health	Yes	[7]

**Table 2 cells-14-00595-t002:** Postbiotics overview.

Compounds	Main Function(s)
Short-Chain Fatty Acids (SCFAs) Acetate, propionate, butyrate	Anti-inflammatory and immunomodulatory effects; glucose and lipid metabolism regulation; energy production contribution
Lactic Acid	Essential for glycolysis in hypoxia or anaerobic conditions; healthy vaginal environment maintenance
Bacteriocins	Immunomodulatory functions; antimicrobial activity
Secondary Bile Acids Deoxycholic Acid (DCA), Lithocholic Acid (LCA)	Dietary lipid adsorption facilitation; anti-inflammatory and immunomodulatory activity, apoptosis signaling modulation [15]; anti-aging benefits [16]; tight junctions stability maintenance [17]
Bacterial Cell Wall Components Exopolysaccharides (EPS), Peptidoglycan (PG), Lipoteichoic Acid (LTA)	Anti-inflammatory activity; SCFA synthesis upregulation; wound healing and tissue regeneration promotion; ion homeostasis maintenance; immune system response enhancement
Plasmalogens (Pls)	Anti-inflammatory and antioxidant activities; adipogenesis regulation; cognitive function improvement
Intestinal bacteria-derived vitamins Water-soluble (B1, B2, B5, B7, B9, B12), fat-soluble (K)	Immunomodulatory and antioxidant activity; Calcium homeostasis and bone health maintenance; blood clotting regulation
Tryptophan metabolites	Neurotransmitters level regulation and brain health support; anti-inflammatory and antimicrobial activity; intestinal epithelial barrier function enhancement; insulin resistance and lipid metabolism promotion
Conjugated Linoleic Acids (CLAs)	Anti-breast cancer, anti-inflammatory and immunomodulatory activity; lipolysis potentiation; atherosclerosis inhibition; osteoporosis prevention
Polyamines	Intestinal epithelial barrier integrity and epithelial renewal maintenance; immunomodulatory activity; B cell senescence reversion
Phenolic compounds	Anti-inflammatory and antioxidant properties; neuroprotective and cardioprotective effects
Hydrogen Peroxide (H_2_O_2_)	Antimicrobial activity (colonization resistance)
Organic Acids (OAs)	Antimicrobial activity (colonization resistance)
Glutathione (GSH)	Antioxidant, anti-inflammatory and immunomodulatory activity; bone health maintenance
Microbial enzymes Proteases, lipases	Food digestion and adsorption facilitation; anti-inflammatory, immunomodulatory, antimicrobial and anticancer activity; lipid metabolism modulation

**Table 3 cells-14-00595-t003:** SCFAs’ receptors and main functions.

Receptor	Cellular Expression	Function(s)
FFAR3 (GPR41)	Colonocytes	Sensor for luminal SCFAs [42]
	Pancreatic β-cell	GSIS regulation [43,44]
	Sympathetic ganglia	Heart rate and energy expenditure increase [45]
	DCs	Quiescent/tolerogenic DC induction [41,46,47]
	Enterocytes	TJ enhancement and intestinal barrier integrity maintenance [48], NF-κB pathway inhibition [38]
	Goblet cells	Mucus secretion [49]
FFAR2 (GPR43)	Enteroendocrine L cells	GLP-1 release [50]
	WAT	Reduction of lipolysis and fat accumulation [51]
	Pancreatic β-cell	GSIS regulation [43]
	Neutrophils	Chemotactic effect and neutrophil activation (phagocytic activity and ROS formation) [52,53]
	T_reg_ cells	Maintenance of intestinal immune homeostasis [54]
	DCs	Quiescent/tolerogenic DC induction [41,46,47]
	Enterocytes	TJ enhancement and intestinal barrier integrity maintenance [48], NF-κB pathway inhibition [38]
	Goblet cells	Mucus secretion [49]
HCAR2 (GPR109A)	Enterocytes	TJ enhancement and intestinal barrier integrity maintenance [48], NF-κB pathway inhibition [38]
	Adipocytes	Metabolic sensor for lipolysis suppression during starvation [55]
	Neutrophils	Apoptosis induction [56]
	DCs	Quiescence/tolerogenic DC induction [41,46,47], IL-10 secretion [57]
	Colonocytes	Tumor suppressor [58]

DCs: dendritic cells; FFAR: free fatty acid receptor; GLP-1: glucagon-like peptide 1; GPR: G protein-coupled receptor; GSIS: glucose-stimulated insulin secretion; HCAR: hydroxycarboxylic receptor; NF-κB: nuclear factor kappa-light-chain-enhancer of activated B cells; ROS: reactive oxygen species; TJs: tight junctions; WAT: white adipose tissue.

**Table 4 cells-14-00595-t004:** Some of the most common bacteriocins secreted by Gram-positive (GPB) and Gram-negative (GNB) bacteria.

Bacteriocin	Producer	Mechanism(s) of Action
Nisin A	*Lactococcus lactis*	Murein synthesis inhibition [78]; induction of preferential apoptosis and cell cycle arrest, reduction of cell proliferation [79]
Epidermin	*Staphylococcus epidermidis*	Murein and WTA synthesis inhibition [80]
Gallidermin	*Staphylococcus gallinarum*	Staphylococci growth inhibition and biofilm formation prevention [81]
Mersacidin	*Bacillus* sp.	Peptidoglycan synthesis inhibition [82]
Sublacin	*Bacillus subtilis*	DNA, RNA and protein synthesis inhibition [83]
Lysostaphin	*Staphylococcus simulans*	Cell wall lytic enzyme (endopeptidase activity) [84]
Enterocin A	*Enterococcus faecium*	Target cell membrane pore formation [85]
Thiazomycin	*Amycolatopsis fastidiosa*	Protein synthesis inhibitor [86]
Microcin L	*Escherichia coli*	DNA/RNA folding and/or synthesis inhibition [87]
Microcin E429	*Klebsiella pneoumoniae*	Target cell membrane pore formation [88]

WTA: wall teichoic acid.

**Table 5 cells-14-00595-t005:** Distribution and function of secondary-bile-acid-activated receptors in immune cells.

Receptor	Immune Cell Distribution	Function(s)	Reference
GPBAR1	Monocytes/macrophages	Downregulation of inflammatory cytokines (TNFα, IFNγ, IL-6, IL-1β) and upregulation of anti-inflammatory ones (IL-10); downregulation of CCL2, subsequent suppression of macrophages migration and polarization facilitation toward the M2 anti-inflammatory phenotype; inhibition of NLRP3 inflammasome activation	[226,227,228,229,230,231,232]
	KCs	Inhibition of LPS-induced cytokine expression via cAMP-dependent pathways	[231,233]
	DCs	Inhibition of NF-κB pro-inflammatory cytokines; induction of a tolerogenic/quiescent state; apoptosis promotion	[234,235]
	NKT cells	Regulation of type I and II NKT polarization and induction of a tolerogenic phenotype; upregulation of anti-inflammatory cytokines (IL-10)	[236]
FXR	Monocytes/macrophages	Downregulation of inflammatory cytokines (IL-1β, TNFα); inhibition of NLRP3 inflammasome activation; polarization facilitation toward the M2 anti-inflammatory phenotype.	[237,238,239,240,241]
	KCs	Downregulation of inflammatory cytokines (TNFα, IL-6, IL-1β) and upregulation of anti-inflammatory ones (IL-10); downregulation of CCL2.	[242,243]
	DCs	Downregulation of MdCAM-1 in the inflamed site with subsequent retention of DCs in the spleen.	[244]
	ILCs	Regulation of ILC commitment towards functional active ILC2 and ILC3 subtypes.	[245]
	NKT cells	Downregulation of OPN.	[246]
	CD8^+^ T lymphocytes	Regulation of the immune response based on the nutritional status via imitation of the metabolic flexibility of CD8^+^ effector T cells.	[247]
PXR	Monocytes/macrophages	Downregulation of TLR4 signaling; upregulation of inflammatory cytokines (IL-1β) via caspase-1 activation.	[248,249]
	T_h_ lymphocytes	Downregulation of NF-κB and IFNγ.	[250]
	B lymphocytes	Possible attenuation of B1 cell production.	[251]
VDR	Monocytes/macrophages	Upregulation of MKP-1 and subsequent downregulation of inflammatory cytokines (IL-6, TNFα).	[252]
	KCs	Anti-inflammatory effects in liver steatosis; protection against hepatic endoplasmic reticulum stress.	[253,254]
	DCs	Induction of tolerogenic DCs; inhibition of mature DCs; inhibition of TIM4 gene expression.	[255,256,257,258]
	NKT cells	Regulation of iNKT cell development and function.	[259]
	CD8^+^ T lymphocytes	Prevention of CD8^+^ T cell proliferation.	[260]
	T_h_ lymphocytes	Inhibition of T_h_1 cell response.	[261,262]
RORγt	ILCs	Promotion of ILC3 differentiation and function.	[263,264]
	T_h_ lymphocytes	Promotion of T_h_17 differentiation; inhibition of T_reg_ cell differentiation.	[265,266]

CCL2: C-C motif chemokine ligand 2; DCs: dendritic cells; IFNγ: interferon γ; IL: interleukin; ILCs: innate lymphoid cells; KCs: Kupffer cells; LPS: lipopolysaccharide; MAdCAM-1: mucosal vascular addressin cell adhesion molecule 1; MKP-1: MAP kinase phosphatase 1; NF-κB: nuclear factor kappa-light-chain-enhancer of activated B cells; NKT: natural killer T cells; NLRP3: NLR family pyrin domain containing 3; OPN: osteopontin; T_h_: T helper cells; TLR: Toll-like receptor; TNFα: tumor necrosis factor α.

## Data Availability

Not applicable.

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
