# Peer review of "Phenotyping the Chemical Communications of the Intestinal Microbiota and the Host: Secondary Bile Acids as Postbiotics"

_cells, 2025, doi:10.3390/cells14080595_

Round 1
Reviewer 1 Report
Comments and Suggestions for Authors
The authors have written an informative review on ‘postbiotics’ and discussed the terminology, catalogued its various components and provided an overview of their interactions with human physiology and immunology. The reference list is extensive which should be helpful to readers of Cells.
The manuscript is well written where helpful I have given suggestions to improve conciseness and clarity.
* What is the main question addressed by the research?
YES THIS IS A REVIEW AND COVERED A BROAD AREA WITH RELEVANT STUDIES CITED, AND PROVIDED A CLEAR DEFINITION OF THE TERM 'POST-BIOTICS'.
* Do you consider the topic original or relevant to the field? Does it address a specific gap in the field? Please also explain why this is/ is not the case.
YES THE AUTHORS GAVE A DETAILED AND EXHAUSTIVE OVERVIEW OF HUMAN HOST RECEPTORS INVOLVED WITH THE INTERACTIONS OF MICROBIAL PRODUCTS. THIS WILL BE AN INCENTIVE FOR READERS OF THE JOURNAL TO GET INVOLVED IN THIS FIELD.
* What does it add to the subject area compared with other published material?
NOVEL
* What specific improvements should the authors consider regarding the methodology?
NONE
* Are the conclusions consistent with the evidence and arguments presented and do they address the main question posed? Please also explain why this is/is not the case.
YES
* Are the references appropriate?
YES
* Any additional comments on the tables and figures.
FIGURES ARE SUPERB/EXCELLENT AND TABLES WELL LAID OUT
Author Response
REVIEWER 1
The authors have written an informative review on ‘postbiotics’ and discussed the terminology, catalogued its various components and provided an overview of their interactions with human physiology and immunology. The reference list is extensive which should be helpful to readers of Cells.
The manuscript is well written where helpful I have given suggestions to improve conciseness and clarity.
* What is the main question addressed by the research?
YES THIS IS A REVIEW AND COVERED A BROAD AREA WITH RELEVANT STUDIES CITED, AND PROVIDED A CLEAR DEFINITION OF THE TERM 'POST-BIOTICS'.
* Do you consider the topic original or relevant to the field? Does it address a specific gap in the field? Please also explain why this is/ is not the case.
YES THE AUTHORS GAVE A DETAILED AND EXHAUSTIVE OVERVIEW OF HUMAN HOST RECEPTORS INVOLVED WITH THE INTERACTIONS OF MICROBIAL PRODUCTS. THIS WILL BE AN INCENTIVE FOR READERS OF THE JOURNAL TO GET INVOLVED IN THIS FIELD.
* What does it add to the subject area compared with other published material?
NOVEL
* What specific improvements should the authors consider regarding the methodology?
NONE
* Are the conclusions consistent with the evidence and arguments presented and do they address the main question posed? Please also explain why this is/is not the case.
YES
* Are the references appropriate?
YES
* Any additional comments on the tables and figures.
FIGURES ARE SUPERB/EXCELLENT AND TABLES WELL LAID OUT
Thanks for your appreciation as well as for your positive and constructive comments.

Reviewer 2 Report
Comments and Suggestions for Authors
This review dealing with interesting topic is well written and the topic is very interesting.
However, certain statements are not precise or misleading. The different compounds may have positive/negative roles and the negative role of some compounds is completely omitted in the review. For example, the authors are describing benefits of polyamines as essential for intestinal epithelia renewal and other processes but high polyamine levels have been linked to colorectal tumorigenesis. DFMO, or difluoromethylornithine, is a drug that inhibits ornithine decarboxylase, an enzyme involved in polyamine synthesis. In this context, DFMO has shown promise in prevention strategies of colorectal cancer. So high levels of polyamines are not beneficial.
Similarly, hydrogen peroxide may cause significant DNA damage leading to mutations and cancer, however as it reads now it seems that the presence of hydrogen peroxide is beneficial.
Furthermore, there are number of papers describing that secondary bile acids such as deoxycholic acid (DCA) induce production of ROS and link the bile acids to tumorigenesis. Indeed, DCA can induce apoptosis after acute exposure depending on the concentration. However, chronic, long term exposure to deoxycholic acid leads to apoptosis resistance. The combination of apoptosis resistance and DNA damage induced by DCA is a dangerous combination and over time results in cancer development. Importantly, the chronic exposure to low concentrations of DCA is physiologically relevant as bile acids are released in response to high fat diet, which is one of the risk factors for development of certain cancers, for example esophageal adenocarcinoma or colon cancer. The authors can find number of publications on this topic.
Overall, the negative impact of these compounds needs to be discussed in this review. This is in my opinion very important and needs to be described in this review, otherwise it is misleading.
Please, define all abbreviations when they are used for the first time in the manuscript.
Author Response
REVIEWER 2
Consideration 1: This review dealing with interesting topic is well written and the topic is very interesting. However, certain statements are not precise or misleading. The different compounds may have positive/negative roles and the negative role of some compounds is completely omitted in the review. For example, the authors are describing benefits of polyamines as essential for intestinal epithelia renewal and other processes but high polyamine levels have been linked to colorectal tumorigenesis. DFMO, or difluoromethylornithine, is a drug that inhibits ornithine decarboxylase, an enzyme involved in polyamine synthesis. In this context, DFMO has shown promise in prevention strategies of colorectal cancer. So high levels of polyamines are not beneficial. Similarly, hydrogen peroxide may cause significant DNA damage leading to mutations and cancer, however as it reads now it seems that the presence of hydrogen peroxide is beneficial.
Reply 1: Of course, some of the postbiotics described in the present review have negative effects, too. The main aim of this work was to underline the positive effects of such postbiotics, by the way some references describing negative effects of polyamines and H2O2 have been added.
Consideration 2: Furthermore, there are number of papers describing that secondary bile acids such as deoxycholic acid (DCA) induce production of ROS and link the bile acids to tumorigenesis. Indeed, DCA can induce apoptosis after acute exposure depending on the concentration. However, chronic, long-term exposure to deoxycholic acid leads to apoptosis resistance. The combination of apoptosis resistance and DNA damage induced by DCA is a dangerous combination and over time results in cancer development. Importantly, the chronic exposure to low concentrations of DCA is physiologically relevant as bile acids are released in response to a high fat diet, which is one of the risk factors for development of certain cancers, for example esophageal adenocarcinoma or colon cancer. The authors can find number of publications on this topic.
Reply 2: As mentioned before, the major aim of this review is to underline the positive effects of the postbiotics in human health, that is why few references to potential side effects were not added in the manuscript. By the way, some references addressing negative effects of secondary bile acids have been added.
Consideration 3: Please, define all abbreviations when they are used for the first time in the manuscript.
Reply 2: After revising the document, it seems to me that all the abbreviations are specified since the first time they are used in the manuscript.

Reviewer 3 Report
Comments and Suggestions for Authors
This is a very interesting, well-written and well-structured review describing the secondary bile acids produced by the intestinal microbiota positive effects on the host. I think Urbani et al’s work is an enjoyable paper and I think it merits to be published in Cells. I only have a consideration and some minor points like typos and so on to be taken into account before its publication.
Consideration
The authors indicate in lines 46-49 that “metabolites generated during intestinal bacteria fermentation, i.e. secreted proteins and short-chain fatty acids (SCFAs), or molecules representing structural fragments of these bacteria, such as exopolysaccharides (EPS) and lipoteichoic acid (LTA), could be considered as postbiotics” giving the impression that purified metabolites could fit this definition.
However, as indicated in the consensus statement of Salminen et al, 2021 (https://doi.org/10.1038/s41575-021-00440-6), that authors also cite:
“The presence of microbial metabolites or end products of growth on the specified matrix produced during growth and/or fermentation is also anticipated in some postbiotic preparations, although the definition would not include substantially purified metabolites in the absence of cellular biomass”
This statement is clearly summarized in the third point of Box 1 (“Purified microbial metabolites and vaccines are not postbiotics”).
I think this should be somehow mentioned after line 49 to indicate that, even if these metabolites could be actively functional when administered purified, they are not strictly considered as postbiotics under those circumstances.
Minor points
- I find strange the use of “to be borne by” as “to be produced by” instead of “to be carried by” (lines 18, 168, 274, 376). I think it can lead to confusion.
- Section 2.6 (Plasmalogens). As one reads this section, it seems that plasmalogens are only a human-cell component and, therefore, its definition as postbiotic (bacterial component) is compromised. I think authors should introduce their role in bacteria, before they do it in humans.
- Please, substitute “classified a r water soluble” by “classified as water soluble” in line 241.
- Lines 349-351, “This allows to speed up reaction rates reactions that, otherwise, wouldn’t be time-compatible with physiological biological time lines”. I think it reads better with a “,” after “reaction rates” such as ““This allows to speed up reaction rates, reactions that, otherwise, wouldn’t be time-compatible with physiological biological time lines”.
- Line 352: “Microbial enzyme are” I think “enzyme” is plural.
- Line 358: “…not only participate in in protein-rich food digestion”. Please, remove one “in”.
- Line 380: there should be an space in “acids[201]”. This happens several times after this. I have seen the following ones but this list could not be exhaustive: line 381 “acids[202]”, line 388 “FXR[210]”, line 390 “acids[214]”, line 391 “expenditure[215]”, line 393 “intestine[217]”, line 448 “[275]and”, line 452 “applications[213]”, line 453 “antagonists[282]”, line 466 “tissue[288]”, line 492 “span[16]”, line 510 “1[291]”.
- Line 495: “LCA is be able to recapitulate” Please, correct.
- Line 508-509: “Caenorhabditis elegans” Please change to italics.
- Line 533: “will be possible to” I think it lacks the subject, maybe “it” (“it will be possible to”
- Line 534: “These bile acid-microbiota interactions grow could lead to transformative” Maybe is “growth” instead of “grow”? Please, check this sentence.
Author Response
REVIEWER 3
Consideration: The authors indicate in lines 46-49 that “metabolites generated during intestinal bacteria fermentation, i.e. secreted proteins and short-chain fatty acids (SCFAs), or molecules representing structural fragments of these bacteria, such as exopolysaccharides (EPS) and lipoteichoic acid (LTA), could be considered as postbiotics” giving the impression that purified metabolites could fit this definition. However, as indicated in the consensus statement of Salminen et al, 2021 (https://doi.org/10.1038/s41575-021-00440-6), that authors also cite: “The presence of microbial metabolites or end products of growth on the specified matrix produced during growth and/or fermentation is also anticipated in some postbiotic preparations, although the definition would not include substantially purified metabolites in the absence of cellular biomass”. This statement is clearly summarized in the third point of Box 1 (“Purified microbial metabolites and vaccines are not postbiotics”). I think this should be somehow mentioned after line 49 to indicate that, even if these metabolites could be actively functional when administered purified, they are not strictly considered as postbiotics under those circumstances.
Reply: You are perfectly right: I personally found some difficulties in what seemed to be to me a dichotomic definition of the term postbiotic. By the way, I’ll follow your suggestion to overcome this possible misleading definition.
Minor points
Q1: I find strange the use of “to be borne by” as “to be produced by” instead of “to be carried by” (lines 18, 168, 274, 376). I think it can lead to confusion.
A2: Corrections done.
Q2: Section 2.6 (Plasmalogens). As one reads this section, it seems that plasmalogens are only a human-cell component and, therefore, its definition as postbiotic (bacterial component) is compromised. I think authors should introduce their role in bacteria, before they do it in humans.
A2: Thanks for the observation, we didn’t realize such forgetfulness. Actually, it has been quite difficult to find enough literature in support of Plasmalogens function in bacteria. I hope the reference added might be sufficiently exhaustive.
Q3: Please, substitute “classified a r water soluble” by “classified as water soluble” in line 241.
A3: Correction done.
Q4: Lines 349-351, “This allows to speed up reaction rates reactions that, otherwise, wouldn’t be time-compatible with physiological biological time lines”. I think it reads better with a “,” after “reaction rates” such as ““This allows to speed up reaction rates, reactions that, otherwise, wouldn’t be time-compatible with physiological biological time lines”.
A4: Correction done.
Q5: Line 352: “Microbial enzyme are” I think “enzyme” is plural.
A5: Correction done.
Q6: Line 358: “…not only participate in in protein-rich food digestion”. Please, remove one “in”.
A6: Correction done.
Q7: Line 380: there should be an space in “acids[201]”. This happens several times after this. I have seen the following ones but this list could not be exhaustive: line 381 “acids[202]”, line 388 “FXR[210]”, line 390 “acids[214]”, line 391 “expenditure[215]”, line 393 “intestine[217]”, line 448 “[275]and”, line 452 “applications[213]”, line 453 “antagonists[282]”, line 466 “tissue[288]”, line 492 “span[16]”, line 510 “1[291]”.
A7: Corrections done.
Q8: Line 495: “LCA is be able to recapitulate” Please, correct.
A8: Correction done.
Q9: Line 508-509: “Caenorhabditis elegans” Please change to italics.
A9: Correction done.
Q10: Line 533: “will be possible to” I think it lacks the subject, maybe “it” (“it will be possible to”.
A10: Correction done.
Q11: Line 534: “These bile acid-microbiota interactions grow could lead to transformative” Maybe is “growth” instead of “grow”? Please, check this sentence.
A11: Correction done.

Round 2
Reviewer 2 Report
Comments and Suggestions for Authors
The authors indicated that their review was focused on the positive effects of probiotics and they omitted the negative effects. This is not correct, the review should provide balanced analysis of current literature otherwise it is highly misleading. I do not agrre with this. For example, bile acids are clearly associated with development of different GI cancers. They are considered carcinogens (see review Bernstein H, Bernstein C et al. Bile acids as carcinogens in human gastrointestinal cancers. Mutat Res. 2005 Jan;589(1):47-65. doi: 10.1016/j.mrrev.2004.08.001. PMID: 15652226.) There are multiple papers on this topic.
Author Response
We completely understand your point of view: you can find some references added regarding the role of bile acids in carcinogenesis of the gastrointestinal tract.